# Disasters without Borders: The Coronavirus Pandemic, Global Climate Change and the Ascendancy of Gradual Onset Disasters

**DOI:** 10.3390/ijerph18063299

**Published:** 2021-03-23

**Authors:** Katsuya Yamori, James D. Goltz

**Affiliations:** 1Disaster Reductions Systems, Disaster Prevention Research Institute, Kyoto University, Gokasho, Kyoto Prefecture, Uji-City 611-0011, Japan; yamori@drs.dpri.kyoto-u.ac.jp; 2Research Affiliate, Natural Hazards Center 483UCB, Institute for Behavioral Science, University of Colorado, Boulder, CO 80309, USA

**Keywords:** borders, gradual-onset disasters, coronavirus, global climate change

## Abstract

Throughout much of its history, the sociological study of human communities in disaster has been based on events that occur rapidly, are limited in geographic scope, and their management understood as phased stages of response, recovery, mitigation and preparedness. More recent literature has questioned these concepts, arguing that gradual-onset phenomena like droughts, famines and epidemics merit consideration as disasters and that their exclusion has negative consequences for the communities impacted, public policy in terms of urgency and visibility and for the discipline itself as the analytical tools of sociological research are not brought to bear on these events. We agree that gradual-onset disasters merit greater attention from social scientists and in this paper have addressed the two most significant ongoing disasters that are gradual in onset, global in scope and have caused profound impacts on lives, livelihoods, communities and the governments that must cope with their effects. These disasters are the coronavirus pandemic and global climate change both of which include dimensions that challenge the prevailing definition of disaster. We begin with an examination of the foundational work in the sociological study of a disaster that established a conceptual framework based solely on rapidly occurring disasters. Our focus is on several components of the existing framework for defining and studying disasters, which we term “borders.” These borders are temporal, spatial, phasing and positioning, which, in our view, must be reexamined, and to some degree expanded or redefined to accommodate the full range of disasters to which our globalized world is vulnerable. To do so will expand or redefine these borders to incorporate and promote an understanding of significant risks associated with disaster agents that are gradual and potentially catastrophic, global in scope and require international cooperation to manage.

## 1. Introduction

Prior to the current daily news reports on the impacts of the coronavirus pandemic, few Americans would have identified the 1918–1920 “Spanish” flu, a gradual-onset disaster that claimed the lives of 550,000 [1] to 675,000 [2] Americans and caused significant and lasting economic impacts [3] as the worst disaster in American history. Similarly, a resident of Japan might say that the worst disaster suffered in that country was the 1923 Tokyo-Yokohama earthquake, but the risk from the increasing frequency of typhoons and weather-related damage and fatalities due to global climate change will likely outstrip other disasters in damage, disruption and possibly fatalities in the not very distant future. In the broader context of public perceptions, it is not surprising that gradually occurring disasters receive less attention. The rapidly occurring major earthquake, typhoon, flood or fire will dominate headlines in the various news media where one will see vivid images of damage to buildings and infrastructure and read or hear eyewitness accounts of the unfolding disaster and reports of response efforts. The high visibility of these rapidly occurring disasters will inevitably result in calls for changes in public policy, particularly if there is an identifiable failure like a particular type of structure that performed poorly in an earthquake. In contrast, slowly evolving disasters may receive only sporadic media attention or, more likely, attention to rapidly occurring manifestations of an underlying gradual-onset disaster (e.g., a particularly violent storm exacerbated by global climate change). Lest we overstate our case, gradually occurring disasters like the current coronavirus pandemic do indeed receive media and public attention when their effects become acute, but whether they result in lasting policy change and achieve cultural salience is questionable. Thus, gradual-onset disasters may not receive the public policy attention necessary to mitigate the impacts that will inevitably follow.

Paralleling the low visibility of gradual-onset disasters among the public is the lack of adequate attention to these disasters among social scientists, mainly sociologists. The low salience of gradual-onset disasters in sociological disaster research lies in a conceptual framework in which rapidly occurring disasters like earthquakes, hurricanes, wildfires and other hazard events that unfold in seconds to a few hours have served as a paradigm in which disasters are assessed and to which a significant amount of public policy attention is directed. Anthropologists, by contrast, have been more expansive in scope, emphasizing disasters as processes rather than events and thus are less constrained by definitional limitations that have tended to exclude slowly evolving disasters [4,5,6]. Nor do we argue that sociologists alone shoulder the responsibility for policy change in regard to disasters, but as sociologists, we must contribute as other disciplines have in addressing the societal implications of gradually occurring disasters. Further, we recognize that there is diversity within the category of gradual-onset disasters and that global climate change and the current pandemic differ in a number of ways. Nevertheless, we have focused on their commonalities; both would be defined as “nondisasters” according to prevailing concepts in sociology; that is, neither is rapidly occurring, spatially limited, conform to the prevailing phased occurrence of a disaster and disaster response and fall within the bounds of prevailing protocols for disaster management.

In this paper, our objective is to explore the manner in which slowly occurring disasters deviate from this paradigm, drawing on observations from the current coronavirus pandemic and global climate change. We will first explore the history of sociological thought regarding natural disasters and how such events have been defined, and the framework for their study has been established. Next, we will explore how two gradual-onset disasters, the current and ongoing coronavirus pandemic and global climate change have challenged the existing conceptual framework in the context of four “borders”, which are temporal, spatial, phasing and political, or positioning. We will also examine the concept of vulnerability, whether it transcends these borders or whether it too requires modification. Finally, we will discuss the implications for future disaster research and for public policy. As a caveat, both the coronavirus pandemic and global climate change are ongoing and evolving, so the views we express about the specifics of these disasters must be considered tentative.

## 2. Concepts, Frameworks and Borders in Sociological Disaster Research: The Literature

Social science and particularly sociological studies of disaster date back at least one hundred years, but the development of a consistent and conceptually coherent framework that has guided the most focused work can be traced to the late 1940’s, particularly to the establishment of the Disaster Research Center at Ohio State University in 1963 by E. L. Quarantelli, Russell Dynes and J. Eugene Haas. The Disaster Research Center conducted both laboratory and field studies of various types of disasters, challenged the prevailing myths of post-disaster social break down, documented the emergence of community groups that responded to the many needs of the post-disaster community and produced case studies of the major events of the era [7]. Disaster research pioneer Charles Fritz provided one of the earliest definitions of disaster, which delimited the concept as: “an event concentrated in time and space, in which a society, or a relatively self-sufficient subdivision of a society, undergoes severe danger and incurs such losses…that the social structure is disrupted…,” [8] (p. 655). This foundational definition limited the range of inquiry to rapidly occurring hazard events that occur in a limited geographical space and cause disruption to an identifiable social system. Although this definition left open the types of hazard events or processes that could cause disasters, later refinements distinguished between human caused (e.g., war, terrorism, collective violence and climate change), technological (e.g., explosions, nuclear accidents, chemical releases into the atmosphere) and natural disasters (e.g., earthquakes, floods, tornados) [9].

The nearly sixty-year-old definition of a disaster by Fritz begs the question of conceptual clarification during the intervening years and indeed there have been important new ideas, but the temporal and geographical limitations on the concept of a disaster have persisted. In a more recent examination of concepts, Quarantelli and Perry [10] argue against an expansion of the disaster definition to include epidemics (in this case the AIDS epidemic) saying “we are inclined to exclude from the concept of “disaster” all very diffused events, including traditional droughts and famines and certain kinds of epidemics…it is best to think of the concept of a disaster as an occasion involving an immediate crisis or emergency” (p. 335). In justifying a narrow definition of disasters, these same authors suggest that slowly occurring disasters like droughts and famines (and presumably, epidemics) create “murkiness”, which concise definitions of a disaster and empirical generalizations based on such concepts of a disaster are ill equipped to analyze. Quarantelli and Perry further argue that “we should stop trying to squeeze relatively heterogeneous phenomena under one label (which would) improve not only our theoretical understanding of a disaster phenomena, but create knowledge useful for planning and managing purposes.” (p. 334). In summarizing the distinction between rapid and gradual-onset disasters, Dynes [11] states that the existing research is “predominantly Western, community-based, urban, and deals with sudden onset agents from ‘natural’ causes.” Disasters involving slow-onset hazards, in contrast, “involve displaced populations, are predominantly rural, deal with conflict… (and) might represent new, previously unseen types of disaster” (p. 2). Dynes comparison suggests that slow-onset disasters have mainly occurred in the developing world where famines, droughts and epidemics have been more common than in the developed world, but climate change and the coronavirus are global in scope impacting both developing and developed nations and should be addressed with social scientific concepts that expand to incorporate these disasters.

Closely related to the temporal confinement in conceptions of disaster is the spatial limitation or zoning aspect. While our principal focus will be on the questionable geographic confinement of disasters to sub-global regions, we must acknowledge that zoning is one of the most ubiquitous tools of academic disaster social science and disaster countermeasures for multiple hazards [12]. The zones printed on various types of hazard maps, such as red and yellow zone designations for landslides, maps of flood districts around particular rivers, tsunami hazard maps, and fire perimeter maps, are literal expressions of zoning. The various districts created in response to the Fukushima Daiichi nuclear disaster, such as the “difficult-to-return zone” and “evacuation order cancellation preparation zone,” are also, of course, examples of zoning. We may also view geographic designations aimed at disaster prevention and relief, such as the “special zone for reconstruction in response to the Great East Japan earthquake,” “Nankai Trough earthquake disaster prevention districts,” and “tsunami evacuation plan special reinforcement districts”, as examples of zoning more broadly defined. Disaster studies have both affirmed the efficacy of disaster zoning and identified its limits and numerous failures.

Both the efficacy and limitations of zoning to prevent or respond to natural disasters are currently attracting attention from scholars. Research demonstrating efficacy includes a series of studies by Ushiyama [13] and Ushiyama et al. [14], who comprehensively examined the locations where deaths occurred during recent heavy rainfall disasters in Japan. They found that over eighty percent of deaths from flooding and related causes occurred in low-lying areas where flooding was possible due to topographic features, pointing out that “such incidents can by no means be considered ‘unforeseen’ if a geomorphological map is consulted” [13] (p. 76). These same authors concluded that ninety percent of landslide deaths in the 2018 rains occurred in or near landslide hazard areas, and other recent disasters showed a similar locational trend for landslides. These data suggest that measures to reduce storm and flood deaths based on zoning, while not perfect, are reasonably effective. On the other hand, certain facts also suggest a downside to zoning. In the well-known warning by Katada (2012) to “not be misled by expectations,” one of his three principles of tsunami evacuation is one such example. Tsunami inundation zones on hazard maps are naturally uncertain, and if people mistakenly feel safe because they are in a location outside the map’s hazard zone, this can be viewed as a negative result of uncertainties in zoning. In fact, according to Katada, [14] during the Great East Japan Earthquake, 65% of those killed or missing in Kamaishi, Iwate Prefecture, lived outside of tsunami inundation hazard zones. Yamori [15] notes that in many communities—especially those in mountainous regions with challenging topographic conditions—it is impossible to establish a public evacuation site that is not located in a landslide hazard zone, flooding or tsunami hazard zone. Another example is Geller’s [16] observation that most of the earthquakes in Japan since 1979 that have caused 10 or more deaths were in areas designated as lower in seismic vulnerability. Area specification of higher earthquake risk has not been so successful.

In another conceptual development consistent with the preferred narrow definition of a disaster is the “disaster cycle”, a two-dimensional taxonomy that is both temporal and structural [17,18]. The temporal dimension identifies the sequential phases of a disaster as-preparedness, response, recovery and mitigation. Preparedness includes planning and warning; response is conceptualized as both pre-impact mobilization and post-impact emergency response; recovery is divided into early recovery (6 months or less) and restoration (6 months or more), and mitigation consists of actions and attitudes toward and adoption of adjustments designed to eliminate or reduce the impact of a disaster. The structural dimensions, essentially levels of analysis, are ordered in terms of increased structural complexity to include the individual, group, organization, community, society and global or international levels. The disaster cycle has become a standard paradigm for both the academic study of disasters and a template for emergency management planning for multiple rapid-onset disasters [9]. See Figure 1. The disaster cycle framework has found a receptive audience among disaster planners in that specific actions as well as programs of disaster management are assigned to each phase. Training manuals used by all levels of government, disaster-oriented non-governmental organizations and private sector emergency management divisions often begin with this typology.

Recent papers by Staupe-Delgado [19], Hsu [20] and Fiske and Marino [21] that address the concept of disasters have questioned the validity of a definition that limits disaster to rapidly occurring events. All three of these studies argue for a conceptual reconsideration of the temporal aspect of disasters and advocate greater academic and public policy attention to slowly occurring disasters. The Hsu and Staupe-Delgado articles summarize earlier literature, and point to the disadvantages of narrowly defining disaster temporally in terms of both knowledge production and public policy. Hsu’s study is specifically sociological and addresses the need for an expanded typology by addressing slowly occurring or developing disasters. Staupe-Delgado offers a broader multidisciplinary examination of disaster definitions with a focus on rapidly and gradually occurring disasters. The Fiske and Marino study is both a critique of the rapid occurring disaster paradigm and an analysis of climate change as a slowly occurring disaster.

Hsu provides a detailed critique of the Quarantelli and Perry [10] argument that disasters should be narrowly defined and offers some conceptual guidance to a more expanded temporal typology of disaster. Hsu states that the definition of disaster as rapidly occurring has remained unchallenged until recently and that Quarantelli and Perry’s analysis relegated gradual-onset disasters, such as droughts, famines and the incremental spread of disease to consideration as social and ecological problems. The notion of rapidity is relative; in short, disasters that are considered sudden vary in time—while a destructive earthquake will transpire over a few seconds to a few minutes, the approach, warning and landfall of a hurricane will transpire over a period of days. Hsu further argues that the Quarantelli and Perry concept fails to clearly distinguish between disasters as narrowly defined and social and ecological problems in which the earlier analysis classifies slow-onset disasters. The author cites the work of DeMit et al. [22] in justifying a more expanded temporal definition of disaster in that 21st-century disasters have become more complex with impacts that are more protracted in time and space and difficult to manage. Further, following Matthewman [23] and Nixon [24], Hsu points out that disasters are, to a lesser extent, events and, more accurately, processes, which become ongoing adjustments or “slow violence.” Hsu concludes by advocating “a temporal definition of disasters that remains sufficiently complex without being overly open-ended” (p.913). In doing so, he draws on the work of Barton [25], who distinguishes situations of collective stress, which are sudden, gradual and chronic (p. 129). Finally, the temporal critique is extended to the spatial --Hsu argues that there is a need to question the assumption that disasters are necessarily concentrated in space, pointing out that “there is a need to understand how disasters circulate around the world and how they can be spatially diffuse” (p. 915).

Adding to the definitional debate is a paper by Staupe-Delgado, who has noted that “…elusive and slow-onset hazards represent a large part of the global disaster burden (while) conceptual and policy innovations developed by disaster researchers over the last century mainly draw on research focused on sudden-onset disasters” (p. 623). Staupe-Delgado also points out that the assumed phased occurrence of mitigation, preparedness, response and recovery have relatively well-defined beginnings and endings, which may not characterize slow occurring disasters, and specifically from our perspective, pandemics and climate change. The author argues that this more traditional view of disasters provides little insight into the nature and unique challenges of gradual-onset disasters and meager guidance for disaster risk reduction policy and practice. In an extensive literature review focusing on the temporal aspect of disasters, Staupe-Delgado found that the few studies addressing slowly occurring disasters focused mainly on adverse impacts but failed to generalize findings to gradual-onset disasters and their unique challenges. The lack of higher-order generalization has resulted in low visibility and neglect of gradually occurring disasters both empirically and theoretically, under prioritization as a research and public policy priority, and fragmented and delayed response or response to rapidly occurring manifestations of underlying gradual processes. In addition, significant in terms of delayed response is the observation of Wisner et al. [26] that gradual-onset disasters like droughts, famines and epidemics often occur in the least developed regions where they are neglected by developed nations until the disaster has become acute.

Fiske and Marino [21] present a cogent critique of disaster defined as rapid in time and geographically concentrated in their paper on global climate change and public policy. Acknowledging that the prototype for both disaster scholarship and public policy has been the rapidly occurring disaster, they describe climate change as “global, gradual, and cumulative over time, and alters the underlying environmental baselines on which disasters occur” (p. 139). The baseline referred to in this quote has two aspects derived from the nature of slow-onset disasters: first, the underlying gradual environmental changes that characterize global climate change (e.g., higher sea levels, a warmer atmosphere, etc.) exacerbate the incidence and severity of rapid-onset disasters, and second, the invisibility of climate change and the ever-present possibility that belief systems among the public will be altered toward normalization of increased disaster events, allowing decision-makers to continue to regard slow-onset processes as non-disasters. The impacts of slow-onset disasters, which in addition to global climate change include drought, famine and pandemics, are “death and diaspora, loss of property and community, loss of cultural icons and way of life, and loss of livelihoods” (p. 141). These impacts do not fall upon populations equally, but differentially based on social class, ethnicity, demographic category and certain geographic locations. This observation raises the issue of environmental justice and the vulnerability of marginalized groups in the context of gradual-onset disasters.

The political dimension of disaster, including gradual-onset disasters, is taken up in an older but still relevant paper by Olson [26]. Olson first affirms the political nature of disasters saying, “government officials are confronted with the need to not only manage the situation, but to explain it (that is): what happened, why the losses were so high and/or the response so inadequate, and what will happen now?” (p. 155). Thus, disasters increase demands on the political system and decisions must be made at each of the phases of disaster, which, as modified for slow-onset disasters include pre-recognition, recognition, response, recovery and reconstruction. How decisions are made, and the effectiveness of programs can impact the level of approval of elected or appointed officials and in extreme cases the legitimacy of the political leadership. An extreme example often cited by those who have studied the politics of disaster is the fall of the Samoza regime in Nicaragua fueled by a corrupt and ineffective response to the 1972 earthquake. Olson speaks of “agenda control”, which broadly includes the issues, or subset of issues with which governments must deal at a given time. Since not all issues can be addressed, agendas necessarily include a zone or “screen” of non-decision. The author applies these concepts to disasters in declaring that “disasters are political crises because they puncture, at least temporarily, the non-decision-making screens on all the political agendas and thereby place a large number of new, complex and conflictual items on all of the agendas simultaneously – hence the temptation to suppress issues or to define the disaster event in other terms” (p. 162). Agenda control is one aspect of the politics of disaster and the second is what Olson refers to as “constructing meaning, causal stories and blame management” (pp. 162–167), processes that seek to control narratives about the significance of events, their causes, the assignment of responsibility and blame, provide excuses for the occurrence of the event or how it was addressed, and justifications for actions taken or not taken during the course of the disaster.

Closely related to the political dimension of disasters is the expertise that is brought to bear when a disaster occurs, that is, what roles are played by disciplinary experts and how do these experts emerge and claim the authority to define the parameters of the disaster and how well managed is the “hand-off” between those who provide authoritative commentary on the disaster and those who formulate decisions on how the disaster will be managed? Further, a third category of actors must be identified that includes communities either actually or potentially impacted by disaster and the local knowledge that may be consistent with or diverge from that of experts and political decision-makers. The literature in this area is not extensive, but the relationship between these groups can seriously affect the way disasters are understood and the measures taken in response and recovery. More recent work has focused on the importance of local knowledge and engagement by community-based organizations with outside experts and officials in the context of natural and technological hazards [27,28,29]. This emphasis on community engagement in promoting resilience has been a theme of three generations of “frameworks,” the most recent, of which is the Sendai Framework for Disaster Risk Reduction 2015–2030 [30]. While most of the available literature focuses on rapid-onset disasters, we will explore the dynamic of interaction among disciplinary experts, government decision-makers and community-based groups in the context of slowly occurring disasters.

One of the most significant contributions of sociology to the study of disasters is the concept of vulnerability and the many insightful studies that have highlighted the differential impact of disasters, both rapid and gradual based on class, race and ethnicity [31], geographic location and the built environment [32,33], disability [34,35,36] and various demographic categories, particularly gender [37] and age [7,38]. There is a sense in which the concept of vulnerability contributes to our argument that the temporal and spatial aspects of disaster should be broadened because vulnerability is often a product of long-term social processes that preceded disaster, are exacerbated by it and continue to be present when the acute phase or phases of the disaster have passed. Scholarly assessments of disaster vulnerability rarely consider the temporal dimension, and since disaster agents like global climate change trigger rapid-onset disasters and pandemics have acute phases, we must consider all possible factors that produce differential vulnerability. Tierney [7] provides a thorough summary of studies that address disaster vulnerability, which she defines as “a combination of long-term disadvantages, such as those typically associated with race and social class and situational conditions that vary over time and across communities” (p. 126) and include three dimensions: the hazardousness of place, the built environment and infrastructure, and social vulnerability. The main contribution of sociology and the social sciences, in general, has been this last factor. Tierney, citing Fordham [39], emphasizes the importance of “intersectionality” or the fact that vulnerability is an “amalgamation of factors in place and time that dictates that some groups will be harder hit and less able to recover” (p. 128). Intersectionality implies that vulnerability is not an “intrinsic characteristic of members of particular groups” (p. 127) and acknowledges the fact that vulnerability factors are likely to be multiple and cumulative.

In this brief review of the literature, we have called into question the prevailing conceptual framework that disasters are exclusively rapidly occurring events confined in time and space and proceed predictably and reliably from a discreet hazard event to disaster response followed by recovery and an inter-emergency phase of mitigation and preparedness. Drawing on the insights of three recent papers and the literature they summarize, which challenge this conceptual framework, we will provide observations from the two major slow-onset disasters that are currently impacting our world, global climate change and the coronavirus pandemic. These observations will emphasize, in addition to the need for conceptual clarification of disaster, the implications of these two gradually occurring disasters for public policy and vulnerability. It is important to mention that our intention in this paper is not to reject the important work that has been accomplished by social science disaster research historically but to expand the concepts to incorporate the most important disasters of our generation and the generations to come.

## 3. The Dilemma of Gradual Onset Disasters: Discussion

Since the beginning of the 21st century, there have been rapid-onset disasters that have caused devastating impacts. Those, which stand out include the 2004 Indonesia earthquake and Indian Ocean tsunami, the 2005 Hurricane Katrina and the 2011 Great East Japan earthquake and tsunami. These events fell reasonably well within the existing conceptual paradigm of disasters, but the two major disasters that are now ongoing, fall outside the borders of this paradigm and defy easy conceptualization with our current analytical tools are the coronavirus pandemic and global climate change. In this section, we will describe the manner in which current concepts of a disaster fall short in adequately describing and understanding these disasters, which occur in a protracted time frame over months and years and are spatially variable from localized to global. We will point to some aspects of gradually occurring disasters that stretch our understanding of their political management and highlight aspects of a vulnerability that vary from the existing paradigm. Our plan is to examine what is currently known about the coronavirus pandemic and global climate change in the context of four conceptual “borders” in disaster research: the temporal, spatial or zoning, phasing and positioning (status and roles of communities, experts and government decision-makers). We will also address the issues of vulnerability associated with these two slowly occurring disasters.

### 3.1. Temporal Borders

The temporal dimension of disaster as a concept in disaster sociology has been debated over the last 60 years or so, and the notion of rapidity has won out over attempts to broaden the definition and, over this same period, social scientists have documented and analyzed the prevailing disasters of their generation, which were mostly rapidly occurring. However, current and subsequent generations will face global climate change, a slowly evolving disaster, which will require a long-term coordinated global response, will transpire over decades and may result in some level of adaptation rather than “recovery.” One of the downsides of considering slowly occurring disasters as social or environmental problems, as Quarantelli and Perry [10] have argued is that social problems may lack the urgency of disasters and, like racism or economic inequality, which actually are social problems, be treated episodically as the most egregious manifestations of these systemic problems emerge. The episodic nature of the response to global climate change will likely be in the form of response to the most frequent and severe meteorological hazards, the impacts of sea level rise and coastal erosion and other changes that are generated by climate change but occur more rapidly and can be responded to in a more familiar and planned manner. Although global climate change has received considerable attention among “climate” scientists, it has received inadequate attention from social scientists, who could bring the insights of sociology to bear on this issue of existential importance.

Among social scientists who have addressed climate change the recent paper by Fiske and Marino [21] provides several points regarding temporal aspects of global climate change. The prevailing paradigm for scientific understanding of disasters holds that disasters occur rapidly; climate changes are gradual and are “almost imperceptible at any given moment, but lead to permanent changes in the ecology and landscape that will render some homes, communities, and cities uninhabitable” (p. 139). The authors note that there are both public perception and public policy implications of the “invisibility” of global climate change. As the perceived risks of increased and more severe flooding, coastal erosion, and hurricanes (also typhoons in Asia) are accepted by the public over time, they become a new normal, allowing decision-makers to side-step the issues regarding slow-onset processes like climate change as non-disasters.

Like climate change, global pandemics are also slow-onset disasters. According to the World Health Organization (WHO), as of 22 March, 2021, there have been approximately 123 million confirmed cases and 2.7 million fatalities due to the coronavirus pandemic in 223 countries [40]. The exact date of origin in Hubei Province, China, where the coronavirus is believed to have originated, is unknown, but the disease was first reported to the WHO on 31 December 2019, based on cases in Wuhan, Peoples Republic of China. Since that time, the disease has spread throughout the world. Some nations took measures to contain the virus with considerable success (e.g., South Korea, New Zealand and China (including Taiwan)) and others did very poorly (e.g., the United States, Brazil, Italy and India). Successful strategies included isolation of those infected, testing and contact tracing, mask wearing, social distancing, frequent hand washing and shutting down venues where large numbers of people congregate and on a national level, closing borders and restricting travel into and out of the country. Currently, we are 15 months into the pandemic with the resurgence of the virus in regions and nations where precautions have not been rigorously followed. The most analogous previous pandemic disaster was the “Spanish” Flu that began in the fall of 1918 and resurged in three distinct phases until the spring of 1920. The advantage of a century of medical science progress is that the current pandemic is likely to be thwarted by a vaccine that was not possible in the earlier outbreak, and as of this writing, multiple vaccines have been developed, thoroughly vetted and are now available for immunizations on a prioritized basis. The current projection is that multiple independently developed vaccines will be available for general distribution by mid-2021.

One additional aspect of the temporal dimension of disasters deserves mention, and that is that slow-onset disasters are similar to, but not identical with, compound/complex disasters, which may be rapidly occurring but extended by hazards secondary to an initial event. For example, following the 2004 Chuetsu earthquakes, heavy rain triggered landslides extending a disaster that began with a rapid-onset event. The Fukushima Nuclear Power Station accident triggered by the 2011 Tohoku earthquake and tsunami is also categorized as a compound/complex disaster. The two current disasters addressed in this paper present a far greater challenge than those, which are rapidly occurring, confined in space with readily identifiable experts and fall more into the phases of a disaster that are characterized by clearly defined borders with predictable beginning and end states. Having neither established social science concepts to define them nor well-defined plans based on established public policy to combat them, nations are struggling with the long-term prospect of heavy death tolls and social and economic disruption from the pandemic. In the background, while nations struggle to address the pandemic, the relentless evolution of global climate change will continue unabated and manifest itself in more frequent and severe meteorological, atmospheric and hydrological disasters.

### 3.2. Spatial Borders or “Zoning”

Recall that one component of Fritz’s [8] early definition of a disaster was that disasters were events *concentrated in time and space*. Disasters, as narrowly defined, can vary from a relatively small geographic area to multiple nations. Compare, for example, the Great Hanshin-Awaji earthquake of 1995 or the Northridge California earthquake of 1994, which were of limited geographic impact to the 2004 Indian Ocean tsunami that caused casualties and damage in 14 nations. Like rapid-onset disasters, slowly occurring disasters may be confined in space as well. The examples we have cited as slow-onset disasters, droughts, famines and epidemics can be confined to a particular locality or region, but the two major disasters we now confront are global in scope and fall outside the narrow definition established in the early days of social science disaster research. In this section, we will explore the spatial dimension of a disaster as it has been applied and offer some observations and justifications for expanding the definition of a disaster to events, or rather processes, which are global in scope.

The word “pandemic”—the outbreak of a disease affecting many people around the world—is a combination of the Greek words *pan* (all) and *demos* (people). The term itself thus refers to the nullification of spatial borders (universal, to all people in the world). Saeki [41] uses the keyword of political and economic “globalism” to point out that in the case of the coronavirus, this essential characteristic of infectious diseases becomes remarkably apparent. Indeed, even without mentioning the rapid global spread of the virus, the fact that China and the United States—the two countries at the center of the global economy—are, respectively, its point of origin and current epicenter speaks powerfully to the inseparability of the coronavirus pandemic and globalism. Could we not have sealed off the disease before it became a pandemic? Today, that question is merely empty counterfactual thinking. Yet, the fact that the phrase “sealed off” is rooted precisely in zoning-based ways of thinking deserves attention. The risk originates in nature and is caused by humans, while our sense of safety and security is founded in “zoning.” As long as the risk is being controlled through zoning—or rather, as long as it is perceived as being managed in that way—people will not have a strong sense of impending crisis.

This sense of risk confinement through zoning is immediately apparent if we examine the progression of the coronavirus outbreak in Japan. Phrases, such as: “It is something happening in Wuhan,” “It is a domestic problem for China,” “It is limited to certain types of places like houseboats and cruise ships,” and “As long as you do not go downtown…” all indicate that zoning lies at the foundation of the public’s sense of safety and security. The same can be said of those in infectious disease management positions. Protective measures at airports, the prohibition of travel to or from a certain country, requests for citizens to refrain from frequenting business districts, emergency declarations limited to certain well defined regions, and requests for citizens to refrain from visiting certain jurisdictions are all based on the concept of zoning. Above all, the currently trending term “hot spot” or “cluster” is deeply reflective of the zoning concept. However, it appears that under globalization, this societal trump card has lost its power to solve our most difficult problems.

Climate change is inherently global in that the gradual warming of the planet affects the ocean and atmosphere, causing environmental changes that threaten every nation around the world. It is gradual in that the changes occur incrementally and have been taking place since the early industrial era (see Figure 2). It is also cumulative, and many of the environmental changes caused by climate change will be permanent. Though global in scope, its current negative impacts and the secondary disasters it generates are regional and local. Fiske and Marino [21] note that “climate change is unique (among slow-onset disasters) in that it continuously shifts the ecological baselines through subtle and insidious sea rise, warming oceans and land areas, increasing erosion, and declining sea ice and snowpack” (p. 139). In the context of zoning, some nations, regions and localities will experience acute changes. Japan has experienced more frequent and severe meteorological hazards, including rainfall and typhoons, flooding and landslides. California in the U.S. has experienced climate change-related drought and larger, more frequent fires. Eastern U.S. coastal areas and Alaska face sea-level rise, causing erosion, loss of wetlands, and habitat on islands, peninsulas and low-lying coastal lands. Globally, climate change includes impacts on water resources (e.g., water supplies, water quality, irrigation, hydroelectric generation, and fish habitat); agricultural production (e.g., crop yields, infestations, plant diseases, salt water intrusion); loss of forests due to cutting, disease, fire and drought, and impacts on human health due to food scarcity, thermal stress, degradation of air and water quality and vector-borne infectious diseases [42,43].

### 3.3. The Disaster Cycle: Phasing

One of the mainstays of both social science disaster research and disaster planning is the assumption of phased stages of preparedness, response, recovery and mitigation (see Figure 1) as a cycle with more-or-less identifiable beginnings and end states. The second author spent his career in emergency management and planning for various rapid-onset disasters was based on this assumption, and to some significant degree, the agency in which he worked was broken into divisions dedicated to managing and administering these phases. As we noted previously, the assumed phased cycle has served both the social scientist and disaster planner as a template for rapidly occurring disasters like earthquakes, fires, hurricanes and volcanic eruptions. If we are to define disaster more broadly to include slowly occurring disasters, however, the assumption of phased stages becomes opaque and breaks down at several points. We will address this poor fit between the disaster cycle concept and slowly occurring disasters as applied to the coronavirus and global climate change.

Global pandemics like the current coronavirus are rare, though infectious diseases are not, and there have been many epidemics that affected large regions (e.g., Ebola) and specific populations (e.g., HIV-AIDS), and some pandemics of lesser lethality (1957, 1968), but we have not experienced a global pandemic of comparable severity since the 1918–20 “Spanish” flu. The flu pandemic of a century ago occurred in a world that could hardly be labeled “globalized”, but the Great War (World War I) simulated globalized conditions as soldiers moved from country to country, facilitating the spread of disease. Since the existence of viruses was unknown at the time, the only defense was the use of masks, distancing, and isolation, strategies difficult or impossible to implement during wartime or during the repatriation of soldiers and displaced populations in its aftermath. That pandemic persisted for approximately two years and had three surges, or acute phases, the second of which occurred in late 1918 and was the deadliest. We digress on the earlier pandemic because we are in the midst of the coronavirus, and much regarding its persistence is not known. In terms of phasing, which is the subject of this section, we can say the following: despite the precedent of epidemics having occurred in the recent past, warnings of a global pandemic and preparedness in advance of its arrival were few and inadequate; response, while in some nations timely, failed to stop the rapid spread of infections and given the nature of periodic surges, it has never been entirely clear whether we are responding to or attempting to recover from the disease. Perhaps we have witnessed a relatively new phenomenon in disaster evolution that could be termed “response-recovery cycles” in which measures assumed to reduce infections are implemented, appear to be working, and recovery is being achieved only to witness renewed surges in infections, prompting a return to response measures (See Figure 3). Clearly, we have observed ambiguity in the length of response and recovery phases and a lack of clarity as to what phase prevails at any given point.

At the level of public perceptions, recent survey research in Japan revealed anxiety regarding whether respondents could maintain recommended coronavirus response actions, including handwashing, use of alcohol-based disinfectants, proper etiquette when coughing, use of masks and ventilation of indoor spaces *until the situation is resolved*? Repeated administration of the survey over a two-week interval in April 2020 identified a decline in perceived resolve to continue these measures despite a surge in infections over this same two-week period [46,47]. These results reflect growing anxiety over not knowing when the crisis will end or how long preventive measures will be necessary. Another way to describe anxiety felt during the coronavirus pandemic is to say that it arises from being unable to determine which temporal phase of this disaster we are currently in, even though we believe it will end at some point. Are we *right now* at the peak of the calamity, are we *already* entering the recovery phase, or are we *still* merely in the run-up to a long period of suffering? This uncertainty regarding “phasing” is a consistent background note in the coronavirus pandemic. As a result, anxiety over being too late in our response to the coronavirus pandemic coexists with anxiety and concern over being too early. We believe the former concern is clear in criticisms, such as “Why did Japan’s government take so long to declare a state of emergency?” While arguments for a slower response are currently rare, it is quite possible that in the future, critics will point to the negative impacts of the shutdown and reduced social and economic activity resulting from corporate bankruptcies; a rapid rise in unemployment numbers; the collapse of welfare, health, or educational services; devastating impacts on cultural activities; related psychological devastation; and increasing crime, domestic violence, and divorces. If these negative outcomes reach a level comparable to the direct impacts of the virus, then people may begin to question whether certain prefectural emergency declarations should have been issued or whether requests for self-isolation or school closures were made too early. The inability to fully dispel anxiety over unclear phasing—that is, the uncertainty regarding whether actions were taken too soon or too late, which is the defining temporal characteristic of the coronavirus pandemic—is closely linked to problems in the management of natural disasters according to the prevailing “disaster cycle.”

Phasing as applied to global climate change is perhaps more problematic than it is to the pandemic. The impacts of global climate change are profound, universal, and cumulative and pose an existential threat that traditional notions of phased response and recovery fail to address either conceptually or in terms of public policy. Conceptually, climate change has not been readily identified as a disaster per se; rather, it has remained largely uncategorized as an ongoing process that has exacerbated the frequency and severity of rapid-onset disasters. Having no definitional association with a disaster has had the effect of walling off climate change from the analytical tools of social science disaster research and numbing it as an urgent issue of public policy. Being global in scope, it must be addressed in a coordinated international manner, which the Paris (climate change) Agreement was established to facilitate. The accord includes 197 nations with the objective of reducing global greenhouse gas emissions to limit the global temperature increase this century to 2 degrees Celsius above preindustrial levels. The U.S. was a participant in the Paris Agreement until 2017 when the administration of Donald Trump withdrew the nation, denying that climate change was a problem, or even a reality, which obviously made response and mitigation a low priority for the U.S. While it is not defined as a disaster, applying the disaster management paradigm to global climate change is not particularly useful, though there have been efforts to mitigate its effect at a sub-national level in the U.S. Fiske and Marino [21] point out that hazard mitigation policy in the US is poorly adapted to climate change in that it focuses on individual property owners rather than community viability, which climate change mitigation requires.

### 3.4. Disaster Politics: Positioning

Olson [26] observes that “in any disaster, government officials are confronted with the need to not only manage the situation but also to explain it” (p. 154). Since few political officials are disaster subject matter experts, additional actors become involved depending on the type of expertise required to explain the hazard or disaster agent. Sometimes, multiple experts must play roles. For example, if a major earthquake occurs, seismologists will be called upon to provide the magnitude, location, focal mechanism and other parameters of the earthquake. If a particular class of buildings did poorly in the earthquake, structural engineers might be relied upon to diagnose the failure mechanism. In some cases, the required expertise will be unclear or conflicting. A third set of actors are the emergency managers, members of public agencies, who respond to the earthquake and will typically be mobilized under the authority of political officials, who must be at least titular heads of the response and recovery effort. Finally, there is the public that will be either directly impacted by the event or outside of the impact zone, but close enough to be interested and engaged, and through various organizations, including the news media, assume the important role of observers and potential critics of the progress, or lack thereof, in the government’s management of the disaster.

As we have pointed out in previous sections of this paper, slowly occurring disasters are typically defined as something other than disasters and may fall outside the protocols for response to “normal” rapidly occurring events. When slow-onset disasters are recognized, the disaster cycle, as Olsen points out, must be modified with “pre-recognition” replacing “pre-impact” and “post-recognition” replacing “impact” though response, recovery and reconstruction remain the same [26] (p. 156). While we applaud Olson’s recognition that the disaster cycle paradigm must be modified for slow-onset disasters, we feel that response, recovery and reconstruction must be modified as well, particularly the assumption that these phases occur in some systematic manner and have clearly defined beginning and end states. Further, political leaders may or may not acknowledge responsibility for managing and explaining a slow-onset disaster and, to the extent that they do, manage poorly, fail to identify the most appropriate experts, provide inaccurate or inconsistent direction to emergency managers and misinform the public. Unlike most rapidly occurring disasters, slow-onset disasters often involve “tradeoffs” in which preventive measures may have significant negative social and economic consequences making management complex and potentially “no-win” situations for political leaders. Both slow-onset disasters we have dealt with in this paper have required such tradeoffs.

The coronavirus pandemic and global climate change represent complex gradual-onset disasters that have been intensely political in that government officials have struggled, in Olsen’s [26] terms, to explain and manage them. As we pointed out earlier, government officials must rely on experts to explain complex hazards. With some hazards, particularly frequently occurring ones like floods, fires, earthquakes and storms, the knowledgeable experts are easily identified and have typically been relied upon in the past. In many cases, these experts are themselves government officials, who are members of science-oriented agencies and provide ongoing advice to government leaders. Slow-onset disasters like the coronavirus pandemic and global climate change present challenges in explaining these complex hazards in that experts may not be readily identifiable, key elements in understanding the hazards may be unknown and clear pathways to effective response may be unavailable. Such disasters may also be rare or sufficiently complex that expertise requires the contributions of multiple disciplines. Both disasters have been difficult for officials to manage and highlight the fact that painful tradeoffs have been necessary for effective response and mitigation. An adequate response has required major changes in large-scale social and economic processes—isolation at home, closure of school and suspension of business activity for the coronavirus and a large-scale transition from fossil fuel energy generation for global climate change.

The coronavirus has proven to be extremely difficult for many political leaders to both explain and manage. Despite warnings from the World Health Organization and awareness of the lethal nature of the disease in January of 2020, US President Donald Trump downplayed the seriousness of the disease, was slow to mobilize a response and constantly questioned and undermined the advice of his own infectious disease experts. Trump had just weathered an aggressive campaign to remove him from office through impeachment, faced a contentious re-election campaign with the election only months away, and the dilemma of a fast-spreading disease with no vaccine and options that meant social and economic disruption through business closures and self-isolation of individuals and families. His response was to deny responsibility and shift the burden of response to the coronavirus to individual states and local governments. State and local responsibility for coronavirus response in the absence of federal government leadership lacked uniformity, put states in competition for vital medical protective gear and equipment and varied from aggressive measures to almost no measures at all. Consequently, the U.S. has experienced the highest death toll in the world, repeated surges of infection and given both poor government response in some regions and a culture of individualism in which many people defied admonitions to wear masks, remain at home to the extent possible, and avoid large gatherings, the development of a vaccine was seemingly the only option for controlling the pandemic. For cases of the coronavirus in selected countries over time, see Figure 4.

In contrast, many nations, particularly island nations, including Japan, have fared better in dealing with the coronavirus based on national leadership, clear and enforceable mandates regarding the suspension of business and in-person education and have more collectivist cultures that are less inclined to demand the right to defy the advice of experts, exposing themselves and others to infection and possible death. Early in the progression of the coronavirus, a program on NHK Japan featured Nobel Laureate Prof. Shinya Yamanaka of Kyoto University, who was tapped for the program on the coronavirus as a medical expert. However, Yamanaka spoke, not as an expert on infectious diseases, which he was not, but as a layperson, who conveyed a sense of the seriousness of the disease and deferred to those who were true experts. For government leaders and the public, this stepping down from expert status by Yamanaka raises important points regarding the interaction between political leaders and subject matter experts. First, it is not always clear in rare slow-onset disasters as to who constitutes an expert and even those who possess expertise may not have all the information necessary to make definitive analyses and recommendations regarding appropriate courses of action. The importance of appropriate expertise in advising political leaders and the public is also reflected in the fact that rumor and misinformation emerge in ambiguous crisis situations to fill any vacuum of authoritative information.

Like the coronavirus, global climate change presents various dilemmas as a political issue. Fundamentally, climate change has not been defined as a disaster, and though it has not on that account become invisible, it has not been treated with the urgency and resolve that characterize hazards, which are so acknowledged. Thus, for political leaders, climate change and the impacts that it has brought about stand as simply one issue among many and may not be of front burner importance. Because climate change has not, at least yet, caused significant impacts in all regions of most countries, it may appear to political leaders as an issue that can be addressed in less than a holistic manner. Management may also be hampered by the necessary sacrifices and economically unattractive tradeoffs associated with major reductions in carbon emissions and conversion from reliance on fossil fuels to cleaner forms of energy. In many nations, including the United States, protecting the fossil fuels industry has prevailed over an aggressive movement to usher in alternate forms of energy, and the U.S. in 2017 withdrew from the Paris Climate Agreement. For political leaders, identifying experts may also be a challenge since the climate and the changes taking place as a result of global climate change require a multidisciplinary set of specialists. The designation “climate scientist” is nonspecific and, to political leaders, begs the question of who is qualified to make authoritative statements about changes taking place in specific regions and jurisdictions.

Slow-onset events, since they are not regarded as disasters, may be only marginally visible to the public, and their urgency blunted as communities that have not been significantly impacted by them can regard their occurrence as someone else’s problem. Even in regions where the frequency and severity of seasonal or other anticipated rapid-onset disasters occur due to global climate change, the gradual ratcheting up of local disasters becomes the new normal. From an emergency management perspective, the contrast between rapid and gradually occurring disasters is even more stark and consequential. In the United States, rapid-onset disasters, once declared as disasters by governmental authorities are met with a wealth of processes and resources to respond and recover. A “declared” disaster that is beyond the response and recovery capability of a local jurisdiction will trigger the implementation of mutual aid pacts, and resources of unaffected jurisdictions will flow to the disaster area. If a disaster is large enough to receive a federal disaster declaration, federal agencies will provide assistance for recovery. In between disasters, jurisdictions from local to federal develop plans, model potential disasters, conduct exercises and provide disaster education to the public. In the United States, state-level emergency management organizations typically have sections or divisions dedicated to the most frequently occurring disasters with specialists, who manage programs and plans, conduct drills and provide educational programs for local jurisdictions and the public. Slowly occurring “non-disasters” like epidemics fall outside the realm of emergency management and are handled by departments of public health or simply not addressed.

### 3.5. Disaster Victims: Vulnerability

Perhaps the most significant contribution to the study of disasters by social scientists is the detailed understanding of how hazards affect human communities or as Tierney [7] succinctly observes, social scientists, have “focused on disasters, not as physical phenomena, but as social ones” (p. 120). The most consequential bi-product of these analyses over the years has been the knowledge that some categories of people fare significantly worse than others when extreme events occur. While there are many definitions of vulnerability, in our opinion, the most straightforward is that vulnerability consists of “the characteristics of a person or group and their situation that influence their capacity to anticipate, cope with, resist and recover from the impact of a natural hazard” [49] (p.11). The capacity to cope, resist and recover from a disaster varies based on factors, including social class, ethnicity, age and gender, as well as non-social factors of geographic location and the built environment. More specifically, disaster vulnerability is often associated with poor people, women, members of racial, ethnic or religious minorities, the elderly and young children, and people with disabilities or chronic health conditions. Lest we assume that vulnerability is an “inherent or intrinsic” characteristic of these groups, Tierney warns that a more nuanced approach is needed in which status disadvantages in disaster situations intersect, with some more salient than others in the specific contexts of disaster [7]. The question that guides our discussion in this section is whether vulnerability in rapid-onset disasters, which constitute the subjects of the bulk of social science disaster research, differs in significant ways from slowly occurring disasters.

Recall that earlier, we quoted Dynes [11], who, in comparing rapid and gradually occurring disasters, observed that slow-onset disasters “involve displaced populations, are predominantly rural, deal with conflict… (and) might represent new, previously unseen types of disaster” (p. 2). What Dynes had in mind were famines, droughts, epidemics, civil unrest or warfare in predominantly developing nations of the global south. Our emphasis in the quote by Dynes might be better directed at the phrase “and represent new, previously unseen types of disasters.” While global climate change and the coronavirus pandemic are not new or necessarily unforeseen, human-induced climate change is a relatively recent discovery, and a global pandemic had not occurred for a century. Clearly, these gradual-onset disasters are not predominantly rural, nor are they confined to developing nations. Both the coronavirus and climate change affect urban and rural areas and developed and developing nations, are unrelated to warfare or civil unrest, but in the case of climate change, do involve displacement of populations. Keeping in mind Tierney’s caution regarding the intersectionality of vulnerability factors, we will now turn to some examples of differential vulnerability in our two gradual-onset disasters.

Although the coronavirus pandemic is ongoing and ultimate outcomes remain unknown, some studies of social vulnerability have been conducted. Karaye and Horney [50], using data from the US Centers for Disease Control (CDC) and employing quantitative methods tested the effects of socioeconomic status (percentage below poverty, percentage unemployed, per capita income and educational attainment), household composition (age, disability, percentage of single-parent households), minority status (percentage minority and English language proficiency) and housing and transportation (percentages of multi-unit structures, mobile homes, residential density, availability of a vehicle and percentage of group quarters). Overall, they found that racial and ethnic minorities, limited English language ability and lack of a high school level education predicted higher coronavirus case counts. The authors noted, however, significant variations within the U.S. in the salience of these factors. For example, as of May 2020, in the states of Washington and Oregon, minority status, language proficiency, household composition and disability were the most salient factors in coronavirus case counts. In the Gulf Coast states, housing and transportation were more predictive of case counts than minority status and language proficiency. In a separate study focusing specifically on disability, Chakraborty [51] employed data on disability characteristics obtained from the 2018 American Community Survey, which define people with disabilities as members of the civilian non-institutionalized population, who reported having serious self-care, hearing, vision, independent living, ambulatory, and/or cognitive difficulties. The author found that people with disabilities, who are Black, Asian, Hispanic, Native American, below the poverty line, under 18 years of age, and female were more likely to contract the coronavirus than people with disabilities, who are non-Hispanic White, above the poverty level, aged 65 or older, and male, after controlling for spatial clustering.

Vulnerability to global climate change would appear at first glance to be mainly associated with a geographic location as the impacts appear to be related to regions where sea level rise threatens coastal communities or areas where climate change has exacerbated the impacts of frequent rapid-onset disasters related to meteorological and wildfire hazards. However, social scientists have noted that vulnerability is not evenly distributed but is “likely to parse the heaviest damages on the most marginalized areas and people, often those who live in low lying delta areas, on small islands, at high altitudes, and in high latitudes” [21] (p.142) [52]. While location is clearly a factor, those who dwell in locations of high environmental hazards are frequently poor, stigmatized or marginalized groups forced to live in hazardous locations as a result of “colonization, housing segregation, forced relocation, isolation and enclosure” [21] (p. 142). In contrast to rapid-onset disasters that may require short-term evacuations, global climate change has required the far more problematic and socially disruptive process of relocation and resettlement. On this point, Oliver–Smith [53] states that “uprooted people generally face the daunting task of rebuilding not only personal lives, but also communities—those relationships, networks and structures that support people as individuals” (p. 124). The disadvantages experienced in resettlement may involve “homelessness, unemployment, marginalization, the loss of neighborhood and community, mental and physical health challenges, and powerlessness” (p.127). These relocation disadvantages are likely to be experienced by people already challenged by historic hardships imposed by class, race or ethnic identity.

## 4. Conclusions

In the formative years in which disaster sociology was being established, founders like Fritz and Quarantelli were probably justified in conceptually delimiting the field as a means of carving out a niche in an established discipline, which did not address disasters. We feel that it is now time to expand the paradigm to incorporate gradual-onset disasters, the disasters that unfold over periods of months and years and include droughts, famines, epidemics, as well as the current coronavirus pandemic and global climate change. These disasters have claimed millions of lives, destroyed livelihoods, disrupted national economies and required large-scale population displacements. These slow-onset disasters, like those that occur rapidly, have caused extensive, cumulative and permanent damage and imposed significant disadvantages on already socially marginalized groups. The exclusion of gradual-onset disasters from the field of sociology has had consequences in terms of the public visibility of these events, the urgency of public policy to address them and the lack of adequate international frameworks to assure cooperation for those that are global in scope.

We have argued that gradual-onset disasters challenge the conceptual “borders” imposed on the study of disaster and attempted to demonstrate how these borders must be expanded to accommodate a broader conception of disaster. These borders are temporal, spatial, phasing or zoning and positioning. Concepts of social vulnerability, too, must account for a broader set of disadvantages imposed by gradual-onset disasters. We have drawn examples from the two current, gradually evolving disasters that confront our nations and the world. The temporal border must be expanded beyond the parameters of minutes to hours to incorporate disasters that unfold over months, years and even decades. The coronavirus has persisted for over a year at this point, and we know that earlier epidemics have encompassed multiple years. Global climate change has been a process that has spanned more than a century, is cumulative in its effects and may persist for decades to come. Spatial borders must be modified as disasters have become multi-national and global, as exemplified by our two ongoing slowly occurring disasters. The standard sociological model for both the study of disasters and the plans that are intended to address them assume that disasters occur rapidly and pass through phases of response, recovery, mitigation and preparedness and have relatively fixed beginning and end states. Some slowly occurring disasters, particularly epidemics, may display recurring response/recovery cycles that are protracted and persist until treatments are available to end the cycles.

Positioning, which reflects the management of disasters and the roles of various stakeholders, must also be considered in the context of gradually occurring disasters. As we have pointed out, some gradual-onset disasters pose important dilemmas for government leaders due to the difficult tradeoffs necessary to address pandemics and climate change. For many rapid-onset disasters, legal and administrative procedures honed over the course of many disasters and coded into law, facilitate a relatively smooth and efficient response and recovery aided by consensus that these processes and procedures are appropriate. Slowly occurring disasters, particularly those that are rare, lack the legally established foundation for response and recovery and require painful tradeoffs that generate dissensus and conflict, will be particularly difficult to manage. Finally, the concept of vulnerability, so central to sociological analysis of disasters, is perhaps the border with the least requirement for modification to accommodate the addition of gradual-onset disasters into the lexicon of research. While the salience of poverty, race, age, gender, disability and other factors that confer disadvantages vary according to the hazard, these factors also exacerbate the effects of gradual-onset disasters as well as those that occur rapidly. Global climate change may impose the additional burdens of permanent dislocation, resettlement and destruction of habitat to those imposed socially.

In making this argument for expanding the conceptual borders of disaster social science, we join the voices of others who perceive the serious consequences of excluding gradual-onset disasters from the framework for understanding and addressing the pressing issues brought about by these phenomena. This is not to say that social science alone, or the conceptual changes we have advocated, will make these disasters infinitely more manageable, but they will improve their visibility and perhaps stimulate public policies that improve adaptation and formalize mitigation strategies that will more effectively address their impacts on our communities. The hazards we now face are increasingly global, deadly, destructive and pose ever greater challenges to our resilience as nations and as a global community. We believe it essential to develop the vocabulary to understand them as a necessary step in our collective capacity to effectively respond to and recover from them.

## Figures and Tables

**Figure 1 ijerph-18-03299-f001:**
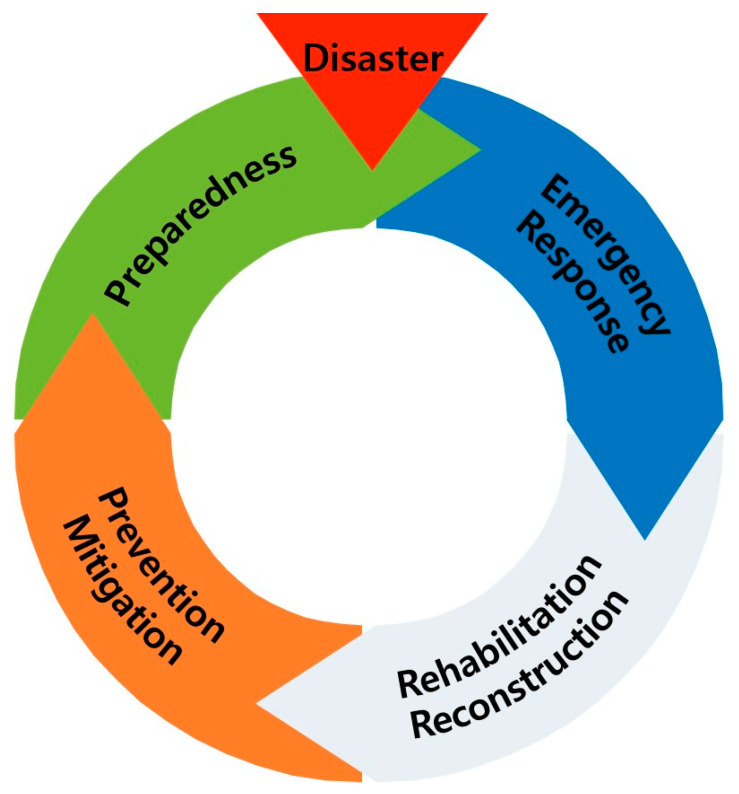
The four phases of disaster (disaster cycle). Source: Hyejeong Park, used with permission.

**Figure 2 ijerph-18-03299-f002:**
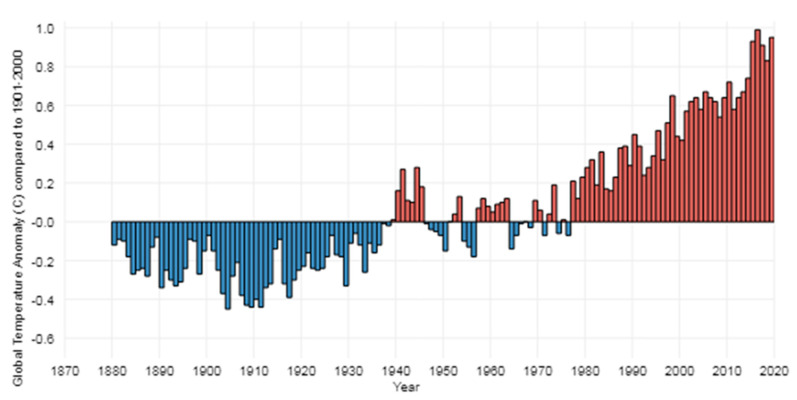
History of global surface temperature since 1880. The five warmest years in the 1880–2019 graph have all occurred since 2015; nine of the 10 warmest years have occurred since 2005. Source: NOAA Climate.gov (authors Rebecca Lindsey and LuAnn Dahlman) [44].

**Figure 3 ijerph-18-03299-f003:**
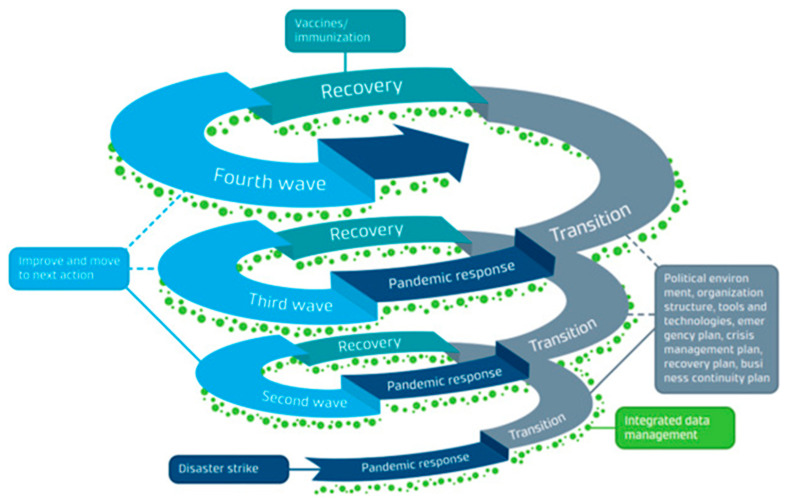
Complex nature of a pandemic management cycle involves “transitioning from pandemic response to recovery in a spiral fashion” [45]. Reproduced unmodified with permission of the author.

**Figure 4 ijerph-18-03299-f004:**
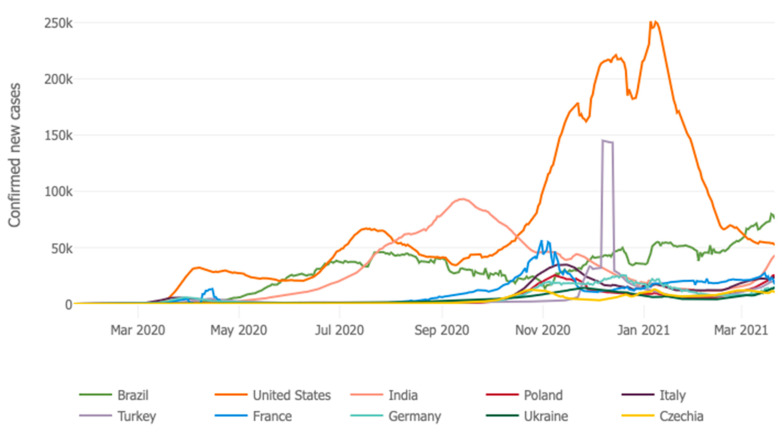
Confirmed cases of the coronavirus in selected nations over time. Source: Johns Hopkins University, Coronavirus Resource Center [48].

## Data Availability

Not applicable.

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
