# Peer review of "Disasters without Borders: The Coronavirus Pandemic, Global Climate Change and the Ascendancy of Gradual Onset Disasters"

_ijerph, 2021, doi:10.3390/ijerph18063299_

Round 1
Reviewer 1 Report
Distinguished from lots of empirical papers on COVID-19 issues recently published, this paper is very philosophical and insightful by proposing unclear “borders” inherent in the emerging types of disasters like pandemics and climate change. This study raises issues against our traditional assumptions on or concepts related to disasters. Therefore, the type of this work should be a “review” article rather than a “research” article, if possible.
Author Response
Response to Reviewer#1: We struggled with how the paper should be classified and decided against classification as a review. Moher et al., 2015 define a systematic review as one that “attempts to collate all relevant evidence that fits pre-specified eligibility criteria to answer a specific research question. It uses explicit, systematic methods to minimize bias in the identification, selection, synthesis, and summary of studies…,the key characteristics of a systematic review are (a) a clearly stated set of objectives with an explicit, reproducible methodology; (b) a systematic search that attempts to identify all studies that would meet the eligibility criteria; (c) an assessment of the validity of the findings of the included studies (e.g., assessment of risk of bias and confidence in cumulative estimates); and (d) systematic presentation, and synthesis, of the characteristics and findings of the included studies.” Our paper is conceptual and theoretical and does not conform very well to the criteria for a review, as defined above under PRISMA-P. As we are new to the IJERPH, we will defer to the editor as to how our article will be classified.
Reviewer 2 Report
REVIEW - "Disasters without Borders: The Coronavirus Pandemic, Global Climate Change and the Ascendancy of Gradual Onset Disasters"
This article is a very good one which makes a good reflexion about disaster concepts and emphasizes a better conceptualization about slow/gradual onset disasters. Only two observations:
Pages 3/4, paragraph: That is a very interesting reflection about the zoning studies results and efficiency. We need to remember that although there are strong evolution about zoning and spatial studies, cartographies and data nowadays, the results are still being generalized and we are seeing the patterns about results (thereby, policies actions will follow these results that are showing patterns and generalization often). In my opinion, this is a common issue within GIS studies as well and certainly is a quandary for a better spatial organization inside a very complex modern world.
Page 5, paragraph 1: Be careful. I think that Hsu's reason to argue Quarantelli and Perry has a temporal and even regional aspect that should not be denied. Always remember that Quarantelli and Perry had specifical goals within the past period of the 1960's. Moreover, the extreme events and their impacts are not only being more complex at 21st Century but the Human capacity to describe them is bigger at these times either. In other words, disasters always were complex and our Society is only having a "little more power" to see them better nowadays.
However, I need to show that a dare work of concepts reorganization or even recreation sometimes is necessary for sure.
That's a very good article, in my opinion. Thank you.
Author Response
Response to Reviewer#2: Reviewer 2 has two comments on the paper. The first deals with zoning or the spatial aspect of disaster. If we interpret this comment to mean that zoning is imperfect and evolving, presumably toward greater accuracy and precision, we would agree with reviewer 2. Technologies like GIS have made zoning, where appropriate, more accurate. Our point in this section was not so much to question the need for zones that distinguish between areas of safety vs. danger, but to raise awareness that for some disasters that are global in scope, zoning may be less relevant and, in some situations, dysfunctional. The second point addresses our treatment of Hsu and his critique of Quarantelli and Perry who argue that definitions of disaster should be confined to events that occur in a spatially confined and temporally limited context. Although this point was not entirely clear to us, we are not critical of Hsu and in fact agree with him. In regard to the historical context of the Quarantelli and Perry definition of disaster, we understand and affirm in the conclusion of our paper that a limited concept of disaster was justified when the field of disaster sociology was new, but we must now expand our definition of disaster to incorporate contemporary disaster agents that affect our globalized world. We did not modify our paper based on comments by Reviewer 2.
Reviewer 3 Report
Thank you for the opportunity to review this interesting paper. Overall, I agree the authors' argument that disaster studies among social scientists should go beyond traditional boundaries; however, I think the authors should be more careful about framing their thoughts. I have a few comments for the development of this paper:
- The authors argue in the beginning that the gradually occurring disasters tend to receive less attention from the public because the rapidly occurring hazards such as earthquakes, typhoons, floods or fires are dominating headlines. Then, the authors bring up the spread of coronavirus. I can partially understand the intended meaning of this statement, but I cannot fully agree with it because all I have seen in the media for the past several months was the spread of coronavirus and the pandemic.
- The authors also argue that the low visibility of gradual onset disasters among the public is due to the lack of adequate attention among social scientists, mainly by sociologist, despite the well documented impacts of such disasters by [natural] scientists. Does this mean that it is the role of 'social scientists' or 'sociologists' to increase the visibility of coronavirus and the climate change among the public? This logic does not make sense to me. Moreover, social impacts of natural hazards have been documented by scholars well beyond the boundaries of sociology.
- I generally agree with the authors' idea that the concept of disaster should be more inclusive rather than exclusive. However, it is also clear that different types of disasters have different characteristics. (How) should we address such differences among different types of disasters? Are pandemics and climate change in the same category as other gradual onset disasters such as droughts and famines? I think the authors can go just a little bit further than simply calling for the inclusion of the two disasters into the boundaries of disaster studies.
- A question I had as I read page 8 was that if a society accepts or adjusts to a gradually developed disaster and creates "a new normal," do policy-makers or social scientists still need to find ways to turn the situation around and go back to where they think was good for everyone?
- It would be great if you could discuss why these two gradually onset disasters (pandemics and climate change) are more significant than other gradually onset disasters (perhaps with some numerical data).
Again, thanks for the opportunity to review this interesting paper. Despite my criticism, I enjoyed reading this paper and learned a lot. I hope my comments are helpful for the development of this paper.
Author Response
Comments and Response to Reviewer #3
Thank you for the opportunity to review this interesting paper. Overall, I agree the authors' argument that disaster studies among social scientists should go beyond traditional boundaries; however, I think the authors should be more careful about framing their thoughts. I have a few comments for the development of this paper:
- The authors argue in the beginning that the gradually occurring disasters tend to receive less attention from the public because the rapidly occurring hazards such as earthquakes, typhoons, floods or fires are dominating headlines. Then, the authors bring up the spread of coronavirus. I can partially understand the intended meaning of this statement, but I cannot fully agree with it because all I have seen in the media for the past several months was the spread of coronavirus and the pandemic.
Although the comment does not have a specific page and line referent, we understand that the reviewer is questioning our contention that gradual onset disasters have relatively low visibility to the public compared to rapid onset disasters. We understand and agree with the reviewer and added the caveat that “Lest we overstate our case, gradually occurring disasters like the current coronavirus pandemic do indeed receive media and public attention when their effects become acute, but whether they result in lasting policy change and achieve cultural salience is questionable.” Our point is less about media attention than it is about the overall cultural salience of gradual onset disasters. Consider, for example, that between the occurrence of many rapid onset hazards like earthquakes, tsunamis and wildland fires, government agencies and NGO’s conduct programs of public education, drills and exercises intended to increase public awareness and preparedness for these hazards. We can think of few gradually occurring hazards for which there exist comprehensive programs of public education and preparedness.
- The authors also argue that the low visibility of gradual onset disasters among the public is due to the lack of adequate attention among social scientists, mainly by sociologist, despite the well documented impacts of such disasters by [natural] scientists. Does this mean that it is the role of 'social scientists' or 'sociologists' to increase the visibility of coronavirus and the climate change among the public? This logic does not make sense to me. Moreover, social impacts of natural hazards have been documented by scholars well beyond the boundaries of sociology.
We agree with the reviewer that other disciplines including the natural sciences and other social scientists have indeed addressed gradual onset disasters, as disasters. We mentioned, for example, that anthropologists have not been constrained by the choices of sociologists to limit the concept of disaster to rapidly occurring hazards. And we had no intention of implying that sociologists alone must carry the burden of policy change in regard to the coronavirus pandemic and climate change. But we are sociologists who focus our research on disasters and feel that our discipline has arbitrarily limited the concept of disaster, has relegated gradually occurring hazards to the realm of social problems and failed to bring to bear the insights gained by the study of rapidly occurring events on those that occur gradually. In recognition of this reviewers comment, we inserted the following sentence in the Introduction, “Nor do we argue that sociologists alone shoulder the responsibility for policy change in regard to disasters, but as sociologists we must contribute as other disciplines have in addressing the societal implications of gradually occurring disasters.”
- I generally agree with the authors' idea that the concept of disaster should be more inclusive rather than exclusive. However, it is also clear that different types of disasters have different characteristics. (How) should we address such differences among different types of disasters? Are pandemics and climate change in the same category as other gradual onset disasters such as droughts and famines? I think the authors can go just a little bit further than simply calling for the inclusion of the two disasters into the boundaries of disaster studies.
The reviewer raises an important point regarding the diversity of disasters within the category we have defined as gradual in onset and we would admit that there are major differences between climate change and the current pandemic. Nevertheless, there are commonalities in that both would be defined as “nondisasters” according to prevailing concepts in sociology; that is, neither are rapidly occurring, spatially limited, conform to the prevailing phased occurrence of disaster and disaster response and fall within the bounds of prevailing protocols for disaster management. We have added a further caveat to the sentence inserted above in response to the reviewer’s comment #2.
- A question I had as I read page 8 was that if a society accepts or adjusts to a gradually developed disaster and creates "a new normal," do policy-makers or social scientists still need to find ways to turn the situation around and go back to where they think was good for everyone?
The answer to the reviewer’s question is yes, as the new normal is usually to the disadvantage of some subset of the population. For example, in gradual onset disasters, the impacts may fall on particular segments of the population like communities of color in the pandemic, or those living in coastal areas vulnerable to sea rise due to climate change. Further, both the population subject to disaster impacts and particularly the unaffected may psychologically adjust to a more dangerous “new normal” when changes occur gradually rather than rapidly. So, we would hold that it is indeed the duty of “action research” to halt the disadvantages and reverse the effects, if possible. In many cases this will not be possible; while one cannot undo the damage climate change has caused nor restore the dead to life, changes in public policy regarding health care and work rules could reduce future losses to minority communities that have experienced differential vulnerability during the pandemic. We feel that these points have been made in the sections on vulnerability and in the conclusion.
- It would be great if you could discuss why these two gradually onset disasters (pandemics and climate change) are more significant than other gradually onset disasters (perhaps with some numerical data).
We do not contend that the coronavirus pandemic and global climate change are more significant than other gradual onset disasters, rather that they are current examples of gradual onset disasters with which most readers of this paper will be familiar. The 1918 pandemic flu, which we do discuss, claimed the lives of at least 50 million people globally compared to the current pandemic that has caused approximately 2.6 million fatalities. Global climate change is serious and ongoing and we do not yet know the full extent of actual (as distinct from projected) impacts. We feel that responding to the reviewer’s comment by adding a comparison of gradual onset disasters including statistical data would add unnecessarily to an already lengthy paper without adding much to the main focus.
Reviewer 4 Report
Thank you for an interesting perspective on the slow-onset enduring disasters. I like the new term: disasters without borders.
I think the article would benefit from a few few minor language related changes. I have made a few comments you might like to consider that I feel will improve the article but would suggest a general read through and review.
On line 6 you say "will be correct" - I would caution against suggesting any disaster is the worst disaster or that anyone is correct and rather say it was a catastrophic or severe disaster, as there are many ways to assess a disaster. On the same line I would also change the "overlook.." to say that "the risk from the increasing frequency of typhoons etc.. will likely outstrip"....I think the article would have a stronger start if you deleted all before line 9 and in fact started the paper with: line 9: "in the broader context of public perceptions, it is not surprising...." I think that provides a strong start and the prelude there is actually not needed and distracting.
P2. lines 2-4 - I would not include the examples or if you do just use one - the levy failure in a hurricane.
P2.Line 10 .."even when implications are recognised and publicised in the media, may not receive"... I would delete the middle phrase
P3. I would remove the line saying "we will return to the definitional issues..." it is not needed and seems out of place.
The paragraph that follows is very well written.
I appreciate the experience from Japanese disasters included and think these are a strong point of the paper.
P3. I would chose either Limits or Pitfalls not use both. They are slightly different. If in doubt I would chose limitations.
P5. Line 1 Again I would not include "a topic that we will take up along with ....". It is not necessary in an article - it is more often used in a presentation.
P6.Line 9 - again I would delete "we will return to the Fiske..."
P6. The last paragraph is a very important concept and you clearly define the three important actors, with crucial inclusion of the Sendai Framework.
P7. I appreciate the first para on vulnerability. It is a difficult topic to write about and you have expressed the concepts well.
P7. I appreciate the sentiment of the last sentence of the second paragraph. It is well said.
P8.Second para last line "we will return..." again i would delete.
P8. last para - i would include "contact tracing" rather than jsut "tracing", replace "evolution" with "progress"
P9. First para Line 10 - I would insert "more" into "...by more clearly defined borders with more predictable ...." as those borders are blurred. The last line is using quite emotive words that havent been used through the rest of the paper and I would suggest that more factual less emotive words were more powerful.
P10. Suggest expanding description of Figure 2. to something more specific in the graph. I would describe the changes in the graph more specifically or I would not use it. It needs to add value to the article (as Figure 3 does brilliantly).
P11. Appreciate the diagramming in Figure 3!
P14. Fig 4 may benefit from labelling the lines as if the article is not published in colour it will not be readable.
I appreciate the significant and original concepts in your article and look forward to quoting the article in my future publications.
All the best.
Author Response
Comments and Response to Reviewer #4
Thank you for an interesting perspective on the slow-onset enduring disasters. I like the new term: disasters without borders.
I think the article would benefit from a few few minor language related changes. I have made a few comments you might like to consider that I feel will improve the article but would suggest a general read through and review.
On line 6 you say "will be correct" - I would caution against suggesting any disaster is the worst disaster or that anyone is correct and rather say it was a catastrophic or severe disaster, as there are many ways to assess a disaster. On the same line I would also change the "overlook.." to say that "the risk from the increasing frequency of typhoons etc.. will likely outstrip"....I think the article would have a stronger start if you deleted all before line 9 and in fact started the paper with: line 9: "in the broader context of public perceptions, it is not surprising...." I think that provides a strong start and the prelude there is actually not needed and distracting.
Response to Reviewer#4: We agree with the changes in wording suggested by the reviewer and the sentence addressing disasters in the US and Japan has been changed. We have not deleted the opening sentences as suggested by the reviewer as we included those sentences to illustrate the low public visibility of gradual onset disasters, both historically and in the present.
P2. lines 2-4 - I would not include the examples or if you do just use one - the levy failure in a hurricane.
We deleted references to multiple examples of high visibility failures, as suggested by the reviewer on page 2, lines 2-4.
P2.Line 10 .."even when implications are recognised and publicised in the media, may not receive"... I would delete the middle phrase
The middle phrase was deleted, as suggested.
P3. I would remove the line saying "we will return to the definitional issues..." it is not needed and seems out of place.
This sentence was deleted, as suggested.
The paragraph that follows is very well written.
I appreciate the experience from Japanese disasters included and think these are a strong point of the paper.
P3. I would chose either Limits or Pitfalls not use both. They are slightly different. If in doubt I would chose limitations.
“Pitfalls” was deleted and “limits” was replaced with “limitations” as suggested.
P5. Line 1 Again I would not include "a topic that we will take up along with ....". It is not necessary in an article - it is more often used in a presentation.
The phrase has been deleted according to the reviewer’s suggestion.
P6.Line 9 - again I would delete "we will return to the Fiske..."
The entire sentence was deleted as suggested by the reviewer.
P6. The last paragraph is a very important concept and you clearly define the three important actors, with crucial inclusion of the Sendai Framework.
P7. I appreciate the first para on vulnerability. It is a difficult topic to write about and you have expressed the concepts well.
P7. I appreciate the sentiment of the last sentence of the second paragraph. It is well said.
P8.Second para last line "we will return..." again i would delete.
The entire sentence was deleted as suggested by the reviewer.
P8. last para - i would include "contact tracing" rather than jsut "tracing", replace "evolution" with "progress"
Both suggested changes have been implemented.
P9. First para Line 10 - I would insert "more" into "...by more clearly defined borders with more predictable ...." as those borders are blurred. The last line is using quite emotive words that havent been used through the rest of the paper and I would suggest that more factual less emotive words were more powerful.
“More” was added to acknowledge blurred borders but we left the last line as it was. We disagree that the words chosen in the last two lines of the paragraph were overly emotive and were not modified.
P10. Suggest expanding description of Figure 2. to something more specific in the graph. I would describe the changes in the graph more specifically or I would not use it. It needs to add value to the article (as Figure 3 does brilliantly).
We have added some additional comments to Figure 2 in response to reviewer 2’s suggestion.
P11. Appreciate the diagramming in Figure 3!
P14. Fig 4 may benefit from labelling the lines as if the article is not published in colour it will not be readable.
We are still seeking a high resolution version of this figure from Johns Hopkins University and, depending on whether color figures are an option, will request an appropriate figure.
I appreciate the significant and original concepts in your article and look forward to quoting the article in my future publications.
Round 2
Reviewer 3 Report
I believe that the current manuscript has addressed all my concerns about the previous manuscript. Thank you for your work.